# Technical Note: No impact of alkenone extraction on foraminiferal stable isotope, trace element and boron isotope geochemistry

Jessica G. M. Crumpton-Banks[1*], Thomas Tanner[2*], Ivan Hernández Almeida[2], James W. B. Rae[1], Heather Stoll[2]

[1]School of Earth and Environmental Sciences, University of St Andrews, St Andrews, KY16 9AL, U.K.
[2]Geological Institute, ETH Zürich, 8092 Zürich, Switzerland

*Correspondence to*: Jess Crumpton-Banks (jgmcb@st-andrews.ac.uk) and Thomas Tanner (Thomas.Tanner@erdw.ethz.ch)

*These two authors contributed equally to this project.

**Abstract.** Recent advances in geochemical techniques mean that several robust proxies now exist to determine the past carbonate chemistry of the oceans. Foraminiferal $\delta^{11}B$ and alkenone carbon isotopes allow us to reconstruct sea-surface pH and $pCO_2$ respectively, and the ability to apply both proxies to the same sediment sample would give strongly paired datasets and reduce sample waste. However, no studies to date have examined whether the solvents and extraction techniques used to prepare alkenones for analysis also impact the geochemistry of foraminifera within those sediments. Here we examine six species pairs of planktic foraminifera, with half being taken from non-treated sediments and half being taken from sediments where alkenones have been extracted. We look for visual signs of contrasting preservation and compare analyses of $\delta^{18}O$, $\delta^{13}C$, $\delta^{11}B$ and trace elements (Li, B, Na, Mn, Mg, Sr, and U/Ca). We find no consistent geochemical offset between the treatments, and excellent agreement in $\delta^{11}B$ measurements between them. Our results show that boron isotope reconstructions of pH in foraminifera from alkenone-extracted sediments can be applied with confidence.

## 1 Introduction

Deep ocean sediment cores provide a wealth of proxy systems for reconstructing Earth's past climate and marine environments. However, obtaining sediment cores is laborious and expensive, and the recovered material is limited and under increasing demand for various complementary proxy analyses, at ever increasing resolution. Consequently, it is important to devise efficient strategies for use of limited core material for multiple proxy systems. In this paper, we demonstrate that we can effectively maximise the application of widely applied proxies including (a) high resolution stable isotope analysis on benthic and planktic foraminifera (Zachos et al., 1996; Lisiecki and Raymo, 2005), (b) extraction of lipids such as alkenones and glycerol dialkyl glycerol tetraether (GDGT) lipids for sea surface temperature (SST) reconstruction and for carbon isotopic determination of alkenones as a $pCO_2$ proxy (Pagani, 2002; Schouten et al., 2013), and (c) measurement of foraminiferal trace element chemistry and boron isotopes to reconstruct SST, pH and $pCO_2$ (Nürnberg 1995; Anand et al., 2003; Foster and Rae, 2016; Rae 2018).

Conventionally, analysis of organic proxies has been made on separate subsamples of core material than that used for the analysis of carbonate proxies. However, it would be ideal to obtain proxy information from the same core depth interval, not only to conserve limited core sample but also to improve intercomparison among proxies. In particular, there is benefit to co-sampling for marine carbonate system proxies, including boron isotopes ($\delta^{11}B$) in marine carbonates and photosynthetic carbon isotope fractionation in alkenones ($\epsilon_p$) (Seki et al., 2010; Rae et al., 2021), as each of these proxies has unique limitations. For example, the 10 Myr residence time of boron in seawater presents a challenge for determining absolute ocean pH values on multi-million year timescales, as changes in foraminiferal $\delta^{11}B$ will be a function of both changes in pH and the $\delta^{11}B$ of seawater (Lemarchand et al., 2002; Foster & Rae, 2016), while phytoplankton-based proxies may struggle to capture low-$CO_2$ conditions due to the upregulation of carbon-concentrating mechanisms in these circumstances (Badger et al., 2019, Stoll et al., 2019). Furthermore, each of these proxies solves for only one component of the carbonate system; combining pH and $pCO_2$ offers us the chance to constrain the carbonate system more fully than we would be able to from either proxy alone (Rae et al., 2021).

Here we evaluate a sample protocol which first extracts lipids from freeze-dried sediment cores, and subsequently isolates the coarse (foraminifera) and fine carbonate for stable isotope, trace element, and boron isotope analysis. We assess whether the high temperature solvent extraction used for lipid extraction impacts foraminifera geochemistry, either through leaching or through contamination which is not removed during the cleaning process. We analysed samples of several species of planktic foraminifera which were split into pairs, where half of each sediment sample had been treated for total lipid extraction by solvents (Accelerated Solvent Extraction, ASE), and half were untreated. We examined the preservation of the specimens using scanning electron microscope (SEM), and analysed several geochemical parameters ($\delta^{18}O$, $\delta^{13}C$, trace element ratios and $\delta^{11}B$) to assess whether foraminiferal geochemistry is affected by the solvent extraction.

## 2 Material and methods

Sediment samples (Table 1) were selected from 3 core sites spanning the western equatorial Atlantic, the North Atlantic and the west Tasmania Margin. Sample 154-926B-20H5, 141-145 cm was taken from Ocean Drilling Program (ODP) Site 926 from the Ceara Rise (3°43'N, 42°54'W, 3598 m), with an age of 7.95 Ma from Wilkens et al., 2017). Sample 342-1406B-8H-6, 8-12 cm was taken from Integrated Ocean Drilling Program (IODP) Site U1406, south of Newfoundland (40°21'N, 51°39'W, 3814 m; age 22.43 Ma, van Peer, 2017). Two samples were taken from ODP Site 1168, off the southeast coast of Tasmania (43°37'S, 114°25'E, 2463 m), one from 189-1168A-25X4 50-52 cm and one from 189-1168A-26X4 50-52 cm; rough age estimate of 13.5 Ma from Stickley et al., 2004. Sediment samples were first freeze-dried for 48 hours and then split into 2 parts, with one half being treated to extract alkenones (ASE treated), and both halves subsequently washed with deionized water to process the > 150 μm size fraction for picking.

## 2.1 Solvent extraction of lipids

The sediment was gently disaggregated while still in the plastic bag from the repository before freeze-drying and was not exposed to any other plastic or glass during the whole pre-sieving process. After freeze-drying, the sediment, still inside the plastic bag, was crushed into small grains using a small rubber mallet to homogenize the sample and increase the surface area for extraction. It is not further ground down into a fine powder to preserve the various microfossils. Clearly, there is a tradeoff between the efficiency of alkenone extraction and the preservation of intact microfossils. The method described here does not involve extreme grinding because it tries to avoid destruction of microfossils and has been used successfully in various publications (e.g. Guitián et al. 2019, Tanner et al. 2020, Guitián et al. 2021). This procedure may reduce the exposure of alkenones within the sediment to the organic solvent and hence reduce the extraction efficiency, compared to vigorous grinding with pestle and mortar. However, any variations in extraction efficiency would not change the result of paleoceanographic proxies that are based on the ratios of organic compounds.

Half of the freeze-dried sediment with a dry weight between 18g to 24g, was extracted using a Thermo Dionex 350 accelerated solvent extractor at the Department of Earth Sciences of ETH Zürich. Therefore, the sediment was put in 34ml stainless steel cells and extracted with three 10-minute static cycles at 100°C with a 5:1 ratio of dichloromethane to methanol (DCM/MeOH). The DCM is a biotech grade solvent (602-004-00-3) from Honeywell and the MeOH is a liquid chromatography grade solvent (1.06007.2500) from Merck KGaA. After three cycles, each extraction delivered a total solvent volume between 85ml to 90ml. We are confident that after three cycles, most of the organic material is extracted from these carbonate rich sediments. Working with similar sediment showed that ~90% is extracted with the first cycle and that the second and third cycle extract the remaining ~10%. Similar results have been reported by Auderset et al. 2020. Subsequently, the now treated sediment was sieved with deionized water through a 150 μm sieve and oven dried overnight at 50°C. The target organic compounds, such as alkenones, are not contaminated by any plastics from the repository bags, because plastic derivatives have much shorter retention times and would elute much earlier in a gas chromatography column. Although alkenones have not been measured in this study, in several years of using this procedure (e.g. Guitián et al. 2019, Tanner et al. 2020, Guitián et al. 2021), plastic contamination was never observed in the earlier part of the chromatograms during alkenone analysis.

**Table 1. Sample ID, age, species, treatment and mass picked for geochemical analyses. Note that samples were initially counted and weighed prior to ultrasonic tests and mounting for SEM analysis; further specimens were subsequently picked for *T. trilobus* and *G. miotumida*. Final numbers analysed were determined from light images taken just prior to crushing and are given in square brackets. [1]mbsf = metres below sea floor (m), [2]mcd = metres composite depth (m), [3]CSF-A = Core depth below sea floor-A (m), [4]Stickley et al. (2004), [5]van Peer (2017), [6]Wilkens et al. (2017).**

| Leg | Site | Hole | Core | Section | Interval (cm) | Depth (top, mbsf[1]) | Depth (top, mcd[2]) | Depth (top, CSF-A[3]) | Age (Mya) | Species | Treatment | Number weighed [analysed] | Mass (mg) |
|---|---|---|---|---|---|---|---|---|---|---|---|---|---|
| 189 | 1168 | A | 26X | 4 | 50-52 | 238.8 | 255.14 | | 13.5[4] | *D. venezuelana* | Pre-ASE | 27 [20] | 1.74 |
| 189 | 1168 | A | 26X | 4 | 50-52 | 238.8 | 255.14 | | 13.5[4] | *D. venezuelana* | ASE | 34 [28] | 2.17 |
| 189 | 1168 | A | 25X | 4 | 50-52 | 229.200 | 245.54 | | 13.0[4] | *G. miotumida* | Pre-ASE | 31 [43] | 0.49 |
| 189 | 1168 | A | 25X | 4 | 50-52 | 229.200 | 245.54 | | 13.0[4] | *G. miotumida* | ASE | 53 [54] | 0.61 |
| 189 | 1168 | A | 25X | 4 | 50-52 | 229.200 | 245.54 | | 13.0[4] | *O. universa* | Pre-ASE | 15 [13] | 0.42 |
| 189 | 1168 | A | 25X | 4 | 50-52 | 229.200 | 245.54 | | 13.0[4] | *O. universa* | ASE | 17 [16] | 0.35 |
| 342 | 1406 | B | 8H | 6 | 8-12 | | | 63.04 | 22.43[5] | *D. venezuelana* | Pre-ASE | 85 [83] | 1.72 |
| 342 | 1406 | B | 8H | 6 | 8-12 | | | 63.04 | 22.43[5] | *D. venezuelana* | ASE | 75 [73] | 1.99 |
| 154 | 926 | B | 20H | 5 | 141-145 | 185.41 | 208.1 | | 7.95[6] | *G. menardii* | Pre-ASE | 110 [105] | 1.61 |
| 154 | 926 | B | 20H | 5 | 141-145 | 185.41 | 208.1 | | 7.95[6] | *G. menardii* | ASE | 130 [128] | 2.27 |
| 154 | 926 | B | 20H | 5 | 141-145 | 185.41 | 208.1 | | 7.95[6] | *T. trilobus* | Pre-ASE | 74 [88] | 2.12 |
| 154 | 926 | B | 20H | 5 | 141-145 | 185.41 | 208.1 | | 7.95[6] | *T. trilobus* | ASE | 51 [108] | 2.50 |

## 2.2. Preparation of foraminifera samples

Several planktic foraminifera species from the > 150 µm size fraction were picked across both sediment treatments: from core 926B, *Trilobatus trilobus* and *Globorotalia menardii*, from core 1406B *Dentoglobigerina venezuelana*, and from 1168A, *Dentoglobigerina venezuelana*, *Orbulina universa* and *Globorotalia miotumida*. These include species which secrete a crust (*D. venezuelana*) as well as thinner walled species which consist of ontogenetic calcite without a crust (*T. trilobus, G. miotumida, O. universa*).

Some foraminifera of both *T. trilobus* and *G. menardii* from core 926B exhibited dark pink crystalline growths on their surface (Fig. 1), which were observed in both the pre-ASE and ASE treated samples. Samples were initially weighed, with sample mass falling between 0.35 – 2.50 mg (Table 1). After weighing, further specimens of *T. trilobus* and *G. menardii* were picked, and some specimens were used for the ultrasonication tests. Between 1 – 3 individuals from each sample were mounted for SEM. *O. universa* and *G. miotumida* from core 1168A were sample limited, with less than 1 mg of sample available for each treatment of these species. Prior to analysis, the remaining specimens were imaged using a light microscope and counted, and then gently crushed between 2 glass slides, with the pink crystals in samples from core 926B being removed by hand at this stage, and the samples then homogenized.

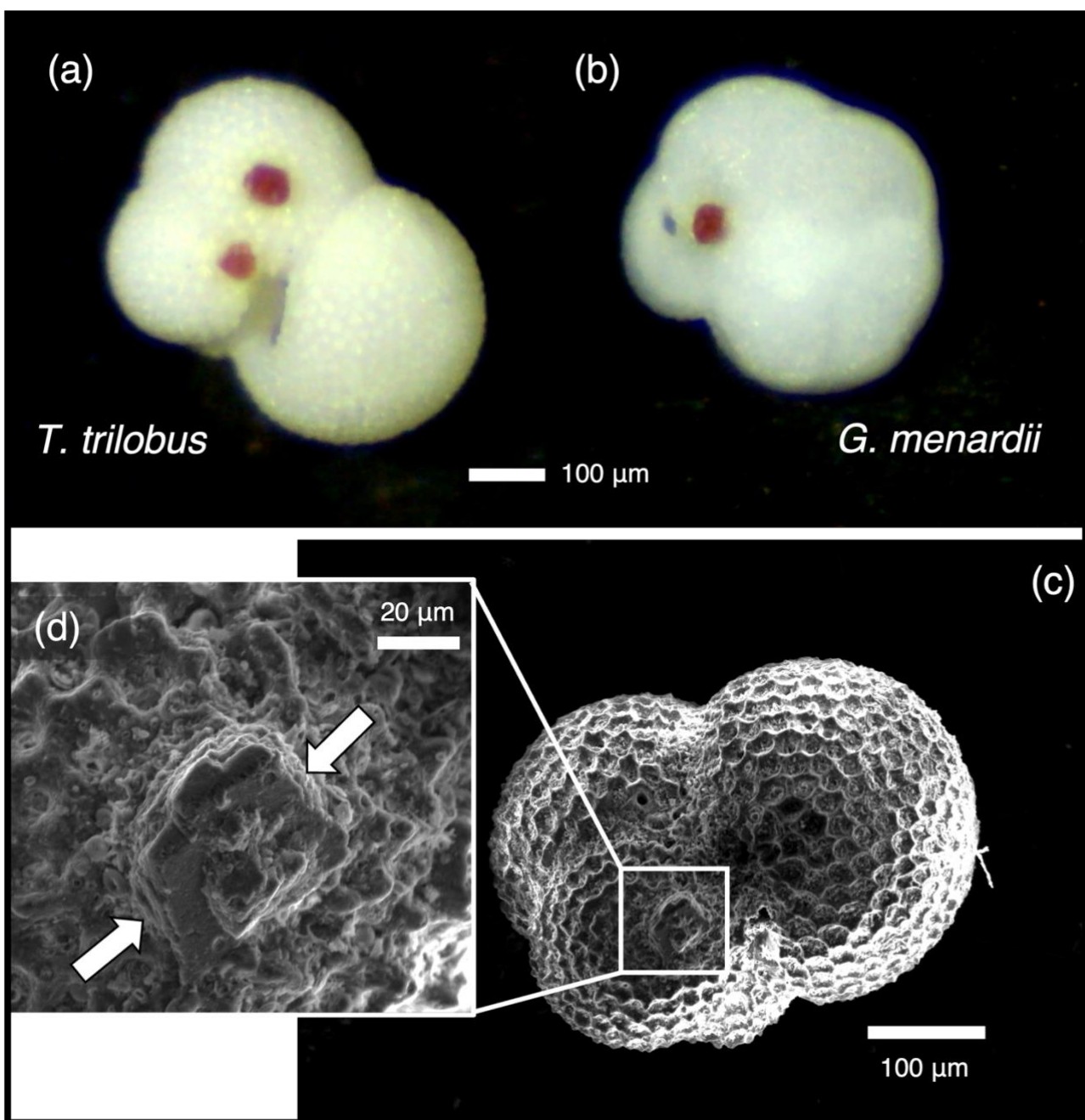

115

Fig 1. Crystal overgrowths were identified on both pre-ASE and ASE samples from ODP 926B, possibly a host phase for the anomalously high Mn (> 400 µmol/mol Mn/Ca) observed in these samples. Shown here on non-treated specimens of (a) *T. trilobus* and (b) *G. menardii.* (c) SEM image of ASE-treated *T. trilobus* with crystalline overgrowth shown in white box and (d) close-up identified by white arrows.

120

121

## 2.2 SEM assessment of preservation

To enable assessment of structural integrity, samples for SEM analysis were gently cracked open using a metal stylus. Fragments were mounted on carbon tape with the broken side facing upwards and were carbon coated. Backscatter electron imaging was carried out on a JSM-IT200 at the University of St Andrews, with accelerating voltages of 10 kV or 15 kV. Visual assessment mainly focused on the cross-section of foraminiferal tests, where the presence or absence of structural features and calcite texture can be strong indicators of overall preservation.

## 2.3 Foraminiferal geochemistry

### 2.3.1 Stable isotopes ($\delta^{13}$C and $\delta^{18}$O)

A small aliquot (~0.2 mg) of the same crushed sample prepared for trace elements and boron isotopes was used for stable isotope analysis. This aliquot was rinsed twice with deionized water and once with MeOH and dried overnight at 50°C. Samples were analysed at ETH Zurich on a GAS BENCH II system coupled to a Delta V Plus irMS (Thermo Scientific) following procedures described by Breitenbach and Bernasconi (2011). Analytical precision after system calibration by two in-house standards and international standards NBS-19 and NBS-18 was 0.14 ‰ for both stable isotopes. Values are reported relative to the VPDB standard.

### 2.3.2 Foraminiferal cleaning

All cleaning prior to dissolution and subsequent sample handling was carried out in class 100 clean facilities at the University of St Andrews. Boron-free MQ water was used throughout cleaning and analysis. $HNO_3$ and HCl acids used were distilled in-house and were of equivalent cleanliness to ultrapure acid. All plastics were subject to acid-cleaning procedures before use. Prior to analysis, foraminiferal samples were mechanically and chemically cleaned to remove clays and organic matter. Mn-Fe oxides may be removed using a reductive clean, but this has been found to notably impact trace element ratios including Mg/Ca (Barker et al., 2003) and so we exclude this step. The cleaning protocol used here follows that of Barker et al. (2003), with some modifications. Initial tests on samples from 1168A found that single specimens *O. universa* and *G. miotumida* were damaged by a few seconds of ultrasonic cleaning and due to this, the ultrasonic time used in each cleaning step was reduced from 30 s to 5 s. The thicker tests of *D. venezuelana* did not show visual signs of damage, but to ensure consistency across samples, all clay-cleaning and oxidation ultrasonic steps were shortened. Al/Ca was measured at < 25 µmol/mol for all samples, indicating that this was an adequate time to remove all clay contamination. Samples were suspended in a small volume (~50 µL) of MQ in a microcentrifuge tube, placed in an ultrasonic bath for 5 s, and clays were removed by adding and then removing ~500 µL MQ. These clay removal steps were repeated for a total of five times. To remove organic contaminants, 250 µL of 1 % $H_2O_2$ solution buffered with 0.1 M $NH_4OH$ was added to the samples, which were then placed in an 80 °C water bath for 5 minutes. Samples were removed from the water bath, opened to release pressure, and then placed in an ultrasonic bath for 5 s.

These steps were repeated three times, with the exception of the smaller mass samples: *O. universa* samples were not given
the oxidative ultrasonication, and *G. miotumida* samples were only given one oxidative ultrasonic step. The oxidative solution
was then diluted with MQ, removed, and the foraminifera fragments were rinsed a further two times. Samples were transferred
to new acid-cleaned vials, and a weak acid leach of 250 µL 0.0005 M $HNO_3$ was applied for 30 s before being removed and
samples being rinsed three times with MQ. Samples were dissolved in 100 µL MQ and 40 µL 0.5 M $HNO_3$, with additional
20 µL aliquots of 0.5 M $HNO_3$ being used to aid dissolution as required. Samples were then transferred into Teflon or plastic
vials and Parafilmed prior to trace element and boron isotope analysis.

### 160 2.3.3 Trace metal analysis

Trace element ratios of the dissolved samples were analysed by triple quadrupole ICP-MS at the University of St Andrews.
Ca, Li, B, Na, Mg, Al, Mn, Sr and U were measured. An in-house trace element standard (BSGS spiked with NIST RM 951)
was used to bracket samples and consistency standards. A small (3 µL) aliquot of sample was diluted and analysed for Ca
concentration; these results were used to dilute samples and standards to a consistent [Ca] matrix of 1 mM. In-house
consistency standards CS1, CS2 and CS3, as well as NIST RM 8301F (Stewart et al., 2021), were measured before and during
the run. Reproducibility of 8301F within these analytical sessions (n = 5) at 2SD is as follows: Li/Ca 2.42 % (9.01 µmol/mol),
B/Ca 0.93 % (138.9 µmol/mol), Na/Ca 2.36 % (3.06 mmol/mol), Mg/Ca 1.97 % (2.62 mmol/mol), Al/Ca 2.73 % (90.91
µmol/mol), Mn/Ca 0.68 % (49.40 µmol/mol), Sr/Ca 0.43 % (1.34 mmol/mol) and U/Ca 1.58 % (68.70 nmol/mol). For B/Ca
we also report the uncertainty on CS3 (3.02 %), which is closer in B/Ca (40.70 µmol/mol) to that of the samples (41 – 79
µmol/mol B/Ca). All Al/Ca values fell below 25 µmol/mol, with 9 out of 12 samples having an Al/Ca value less than 10
µmol/mol, indicating that the dissolved samples were clean of clay contamination despite the shorter ultrasonication time
applied.

### 173 2.3.4 Boron isotope analysis

Boron isotopes were analysed at the University of St Andrews on a Neptune Plus MC-ICPMS equipped with $10^{13}$ Ω resistors.
Separation of the sample boron from the carbonate matrix was carried out using columns filled with Amberlite IRA-743 (Kiss,
1988) and closely followed the procedure of Foster (2008). Samples were buffered using an ammonium acetate buffer (1.1 M
ammonium hydroxide:1.2 M acetic acid, exact concentrations adjusted to achieve pH 6) at 1.5x the volume of acid used for
dissolution and eluted in a small volume (9x 50 µL) of 0.5 M $HNO_3$ to boost the measured signal. To aid washout and boost
signal, eluted samples were spiked with Romil-Spa ultrapure HF to a concentration of 0.3 M (Rae et al., 2018; Zeebe & Rae,
2020). Total procedural blanks were small (< 10 pg B, n = 3), with all procedural blank corrections applied being < 0.04 ‰.
Sample size and indicator elements for contamination (Na, Ca, Mg), either from remaining carbonate matrix or other sources,
were assessed prior to analysis, with no signs of remaining matrix or secondary contamination identified. Due to differing
initial starting masses, sample size was variable, with the lower mass samples of *G. miotumida* and *O. universa* falling between
0.4 – 1.2 ng B; the remaining samples had 3.7 – 11.8 ng B. For the larger samples, the sample-standard bracketing approach
of Foster (2008) was used. The smaller samples gave low concentrations (< 1.6 ppb [B]), which can make samples more
sensitive to an inaccurate in-sequence blank correction; for this reason, these samples were individually blank-corrected and
blocks of four samples were standard-bracketed. The average difference between bracketing standards for samples analysed
in this way was < 0.15 ‰, which is comparable to the main run. NIST RM AE121 (main sequence) reproduced at $19.66 \pm 0.17$
‰, and the in-house standard BIG-D reproduced at $14.82 \pm 0.13$ ‰ (main sequence, 10 ppb) and $14.86 \pm 0.36$ ‰ (small sample
sequence, 5 ppb). The procedural standard NIST RM 8301F (Stewart et al., 2021) reproduced at $14.53 \pm 0.08$ (2SD) ‰ (n =
7, 10 ppb). Uncertainty is shown at 2SD equivalent based on the characteristic reproducibility of standards of equivalent size
to the samples run here (0.20 ‰ for the main run samples, 0.40 ‰ for the smaller samples). This is likely conservative given
that samples in the main run were all run at greater concentrations than the full procedural NIST RM 8301F standard, which
was run at 10 ppb and reproduced at 0.08 ‰ (2SD, n=7). Due to sample loss, only 1 replicate of ASE-treated *T. trilobus* was
carried out, and so a greater uncertainty of 0.35 ‰ was assigned to this sample.

## 197 3 Results and discussion

### 198 3.1 Physical preservation

SEM assessment of the foraminifera finds variable preservation between sites and species (Fig. 2), but little difference in
preservation between treatments. The exception to this is in both pairs of samples from 926B, where foraminifera from the
ASE-treated sediments possibly appear slightly better preserved than the non-treated samples. Fine laminations are better
preserved in ASE-treated *G. menardii* specimens from 926B compared to the non-treated specimens, where broken faces show
more signs of etching (a more "ragged" appearance), and in some instances it is difficult to identify the individual layers of
calcite or fine features such as the location of the primary organic membrane. *T. trilobus* individuals from 926B show a similar
pattern of preservation, with better preservation of fine details through the test in the ASE-treated samples (see inset in Fig.
2d), while the non-treated samples appear more heavily etched. We also note the lack of structures such as pores visible on the
internal surface of the ASE *T. trilobus* specimen examined, with an accompanying smooth appearance suggesting the presence
of an authigenic phase. The remaining pairs from cores 1406B and 1168A show consistent preservation patterns within pairs.
*D. venezuelana* specimens from core 1168A exhibits good preservation, with minimal etching, smooth internal test surfaces
and smooth calcite crystals faces in the crust calcite. *O. universa* and *G. miotumida* specimens from 1168A show less good
preservation, with etched calcite and pits visible in pores, though we note that the internal trochospiral part of the *O. universa*
test was intact and visible in both individuals imaged. This discrepancy in preservation between species in the same sample
may be due to the presence of a thick dissolution-resistant calcite crust in *D. venezuelana* (Schiebel & Hemleben, 2017; Petró
et al., 2018). *D. venezuelana* also has low Mg/Ca compared to many of the other species in this study (see section 3.3), which
may further contribute to a higher preservation potential. *D. venezuelana* from core 1406B shows less good preservation than
in core 1168A, with etching and pitting visible, likely due to differences in preservation potential between the sites, alongside
a lower degree of crusting and slightly higher Mg/Ca.


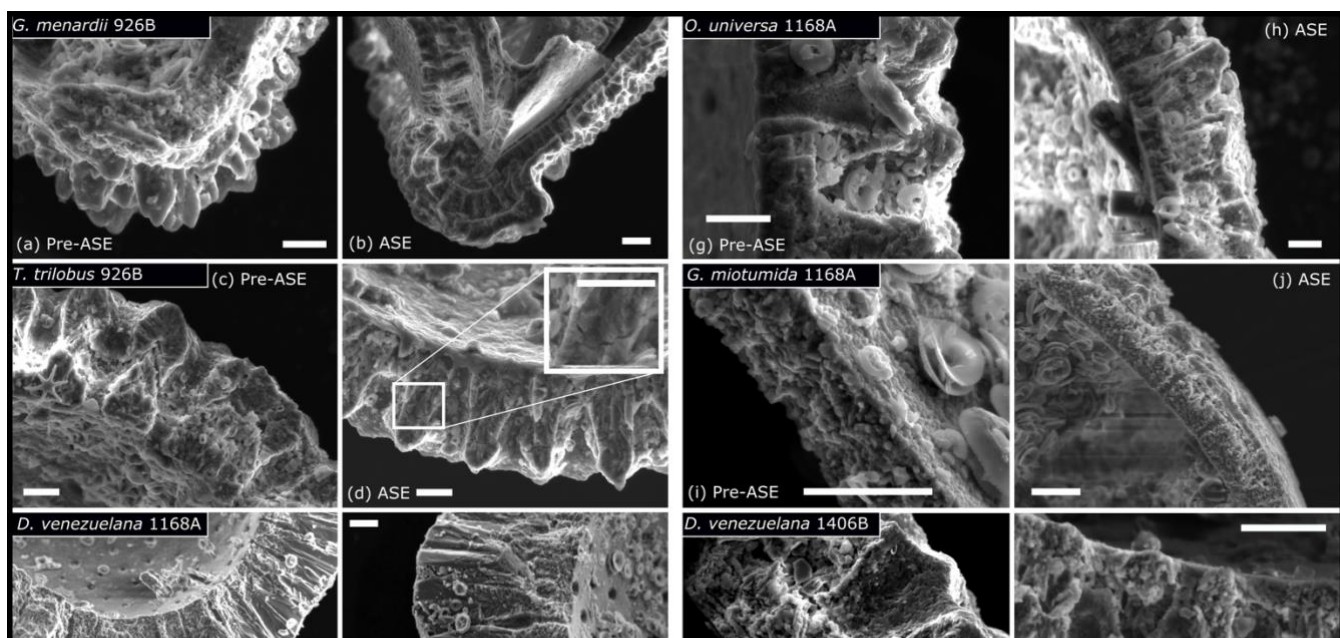


**Fig 2. SEM images assessing impact of ASE treatment on microstructural preservation of foraminifera. a) Pre-ASE *G. menardii***
**from 926B show signs of etching and the loss of fine detail in some samples, while ASE treated specimens (b) have laminations**
**preserved, although etching across the test wall is present. c) Pre-ASE *T. trilobus* from 926B are also etched, with some rough inner**
**surfaces and the loss of fine features between calcite layers, while these appear better preserved in samples from the ASE treatment,**
**with the inset showing a higher magnification view of fine cross-sectional structural details (d); though note the loss of pores on the**
**inner surface of ASE-treated *T. trilobus*, indicating mineral overgrowth. e) *D. venezuelana* from 1168A pre-ASE treated and f) ASE**
**treated. In both the inner surface is smooth with no etching, and the outer calcite crust is well-preserved. Some slight etching of**
**laminar calcite visible. g) *O. universa* from 1168A pre-ASE treated and h) ASE treated. Etching visible in test cross section for both,**
**as well as pitting in pre-ASE pore surface, though inner trochospiral form is preserved in both samples (not shown). i) Pre-ASE**
**treated *G. miotumida* from core 1168A and j) ASE treated. As for the *O. universa*, etching is visible across the test wall. k) *D.***
***venezuelana* from 1406B, pre-ASE treated and l) ASE treated. Etching and pitting visible in both, although fine features such as**
**layers are preserved. All scale bars shown are 10 μm.**

The potential for better preservation following ASE treatment is unexpected. One possible explanation for the slight differing
preservation between treatments observed in sediment core 926B could be sorption of the solvents used in alkenone extraction
to the calcite surface, which might then protect the foraminifera from dissolution due to undersaturation of water used during
sediment washing. We do note that the ASE-treatment will effectively perform an organic clean on the foraminifera; part of
the difficulty in identifying fine features in the pre-ASE *T. trilobus* may therefore be due to the presence of organics and debris
on the broken faces. However, there appears to be slightly more gaping between the calcite layers in the pre-ASE *T. trilobus*,
which seems more likely to reflect preservation than lack of cleaning. We also note that the low number of individuals assessed
by SEM for each species treatment ($< 3$) makes it possible the discrepancy is due to heterogeneity between specimen
preservation. Based on the results of the SEM assessment of samples, ASE treatment does not appear to negatively impact the
preservation of foraminifera hosted in the sediment.

## 244 3.2 Stable isotope results

The analysis of $\delta^{13}C$ and $\delta^{18}O$ across the pairs of treated and non-treated species does not reveal a clear or systematic offset
(Fig. 3, Table 2). The difference in $\delta^{13}C$ between five out of six pairs is $\leq 0.11$ ‰, which falls within 1 SD of each other (with
1 SD being the reported 0.07 ‰ analytical precision). The offset of 0.25 ‰ in the *O. universa* pair is still within 2 SD and is
likely due to the limited sample size (pre-ASE n = 13, ASE n = 16, noting that only a small fraction of this total was used for
stable isotopes following crushing and splitting), rather than any influence of the ASE-treatment (Fig. 3b).

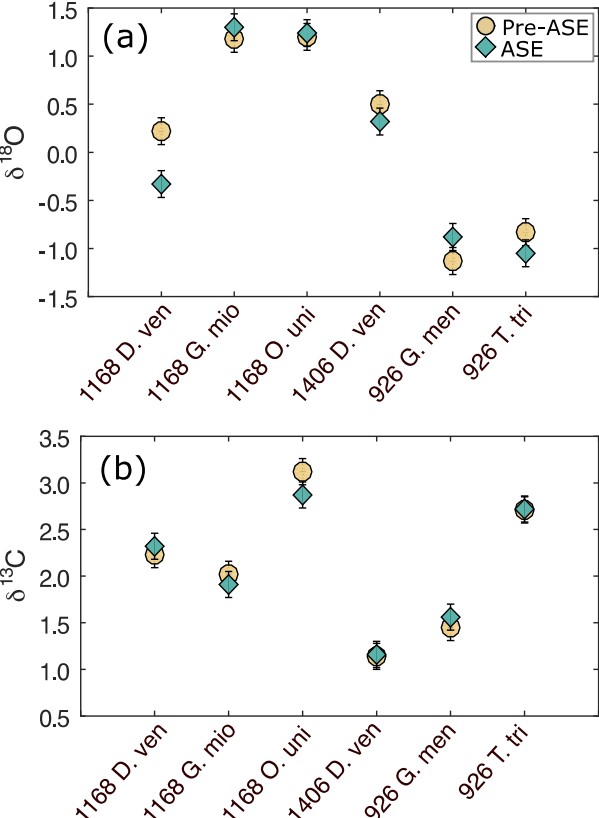

**Fig 3. (a) $\delta^{18}O$ and (b) $\delta^{13}C$ results for the comparison of pre-ASE (yellow circles) and ASE treated (blue diamonds) foraminifera. Uncertainties are shown as analytical precision of 0.14 ‰. There is no consistent offset between species pairs, supporting the use of stable isotopes in foraminifera from ASE treated sediment.**

271

Table 2. $\delta^{18}O$, $\delta^{13}C$ and $\delta^{11}B$ results for ASE-treated and non-ASE samples. DV = *D. venezuelana*, Mi = *G. miotumida*, Ou = *O. universa*, Men = *G. menardii*, Tt = *T. trilobus*.

| Sample | $\delta^{18}O$ | 2SD | $\delta^{13}C$ | 2SD | $\delta^{11}B$ | 2SD |
|---|---|---|---|---|---|---|
| | (‰) | (‰) | (‰) | (‰) | (‰) | (‰) |
| 1168Dv | 0.22 | 0.08 | 2.23 | 0.02 | 12.86 | 0.20 |
| 1168Dv-ASE | -0.33 | 0.06 | 2.32 | 0.06 | 12.98 | 0.20 |
| 1168Mi | 1.18 | 0.10 | 2.02 | 0.02 | 12.70 | 0.40 |
| 1168Mi-ASE | 1.30 | 0.08 | 1.91 | 0.08 | 12.57 | 0.40 |
| 1168Ou | 1.20 | 0.02 | 3.12 | 0.06 | 13.14 | 0.40 |
| 1168Ou-ASE | 1.24 | 0.12 | 2.87 | 0.06 | 13.63 | 0.40 |
| 1406Dv | 0.50 | 0.06 | 1.14 | 0.04 | 14.27 | 0.20 |
| 1406Dv-ASE | 0.32 | 0.08 | 1.16 | 0.08 | 14.38 | 0.20 |
| 926Men | -1.13 | 0.14 | 1.45 | 0.04 | 16.82 | 0.20 |
| 926Men-ASE | -0.88 | 0.14 | 1.56 | 0.04 | 16.87 | 0.20 |
| 926Tt | -0.83 | 0.02 | 2.71 | 0.06 | 17.87 | 0.20 |
| 926Tt-ASE | -1.05 | 0.10 | 2.72 | 0.04 | 17.66 | 0.35 |

The variability among the $\delta^{18}O$ pairs is in general higher than that for $\delta^{13}C$, and three pairs have an offset that is larger than 1 SD. *T. trilobus* from 926B and *D. venezuelana* from 1406B have an offset of 0.22 ‰ and 0.18 ‰, respectively, though still fall within 2 SD of the measurement. The only clear outlier is *D. venezuelana* from 1168A with a difference of 0.55 ‰, which after *O. universa* was the second smallest species pair (pre-ASE n = 20, ASE n = 28; and as for *O. universa,* only a small fraction of this crushed sample was analysed for stable isotopes). Therefore interspecimen variability might explain this offset. A species-specific influence cannot entirely be ruled out but seems unlikely since the second pairing of *D. venezuelana* from 1406B is within 2 SD and consisted of up to 4 times the amount of single specimen picked (pre-ASE n = 83, ASE n = 73). These results support the application of $\delta^{13}C$ and $\delta^{18}O$ in foraminifera from ASE-treated sediments to be used in palaeoceanographic reconstructions.

**3.3 Trace element results**

There is no consistent offset in trace element ratios between the treatments across the pairs studied (Fig. 4, Table 3). Notably, sample pairs with larger numbers of individuals (*T. trilobus*, *G. menardii* and *D. venezuelana* from 1406B) tend to give values within analytical error (see Table 1 for specimen counts). More variability is observed in sample pairs with lower numbers of

individuals for some elements, which we attribute to heterogeneity between individual specimens. Percentage difference is
reported relative to the untreated samples.

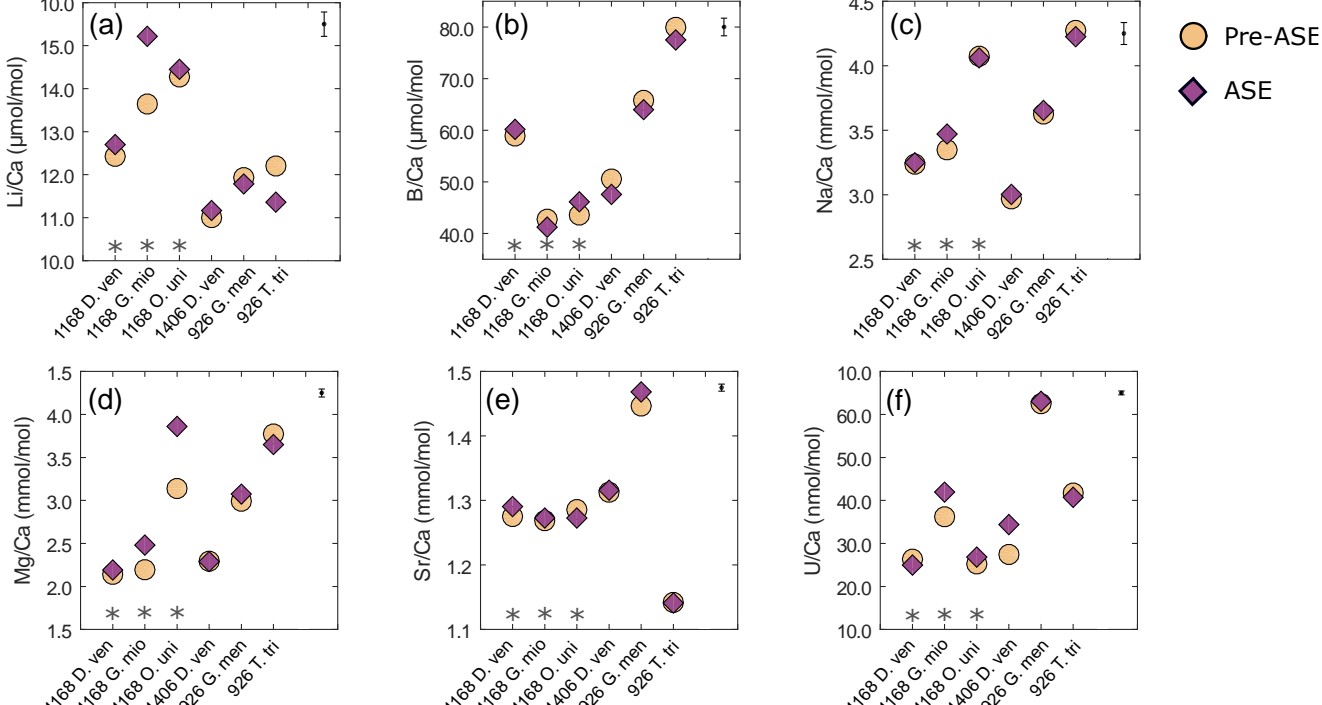

**Fig 4. Results of trace element analyses for pre-ASE (yellow circles) and ASE (purple diamonds) for the species from the core sites**
**studied. Uncertainty on the average sample value for each element is shown in the upper right hand corner. No systematic offset is**
**observed between the treatments. (a) = Li/Ca, (b) = B/Ca, (c) = Na/Ca, (d) = Mg/Ca, (e) = Sr/Ca, (f) = U/Ca. D. ven. =** *D. venezuelana,*
***G. mio =*** *G. miotumida***, O. uni =** *O. universa***, G. men =** *G. menardii* **and T. tri =** *T. trilobus***. Grey asterisks indicate sample pairs**
**where at least one sample has < 70 individuals.**

In Li/Ca, there is a significant positive offset of 12.5 % observed in Li/Ca for *G. miotumida*, likely attributable to the smaller
number of individuals in these samples (pre-ASE n = 43 , ASE n = 54). There is a smaller opposite offset of 7.5 % in Li/Ca
between treatments for *T. trilobus* which is unlikely to be due to sample size (pre-ASE n = 88 , ASE n = 108). However across
the dataset as a whole there is no consistent difference in Li/Ca between treatments. B/Ca ratios were within analytical error
for both treatments (Fig. 4b).  In some cases, small deviations (< 5.9 %) in B/Ca ratios were observed between untreated and
ASE-treated samples, but there was no systematic trend of higher or lower B/Ca with ASE treatment. Na/Ca ratios were all
within uncertainty, with most pairs exhibiting < 1.2 % variability; *G. miotumida* showed a slightly greater difference with 3.7
% difference between the treatments (Fig. 4c).

**Table 3. Trace element results for ASE-treated and non-ASE samples. DV = *D. venezuelana*, Mi = *G. miotumida*, Ou = *O. universa*, Men = *G. menardii*, Tt = *T. trilobus*.**

| Sample | Li/Ca µmol mol$^{-1}$ | B/Ca µmol mol$^{-1}$ | Na/Ca mmol mol$^{-1}$ | Mg/Ca mmol mol$^{-1}$ | Al/Ca µmol mol$^{-1}$ | Mn/Ca µmol mol$^{-1}$ | Sr/Ca mmol mol$^{-1}$ | Cd/Ca nmol mol$^{-1}$ | Ba/Ca µmol mol$^{-1}$ | Nd/Ca µmol mol$^{-1}$ | U/Ca nmol mol$^{-1}$ |
|---|---|---|---|---|---|---|---|---|---|---|---|
| 1168Dv | 11.43 | 58.89 | 3.24 | 1.64 | 5.27 | 70.69 | 1.28 | 47.93 | 6.54 | 0.33 | 16.33 |
| 1168Dv-ASE | 11.70 | 60.17 | 3.25 | 1.69 | 4.91 | 68.50 | 1.29 | 38.59 | 6.72 | 0.32 | 14.97 |
| 1168Mi | 12.64 | 42.76 | 3.35 | 1.69 | 4.51 | 114.19 | 1.27 | 96.01 | 5.88 | 0.64 | 26.21 |
| 1168Mi-ASE | 14.22 | 41.22 | 3.47 | 1.98 | 9.53 | 135.01 | 1.27 | 117.39 | 7.78 | 0.79 | 31.94 |
| 1168Ou | 13.27 | 43.57 | 4.07 | 2.64 | 7.06 | 66.58 | 1.29 | 39.07 | 2.34 | 0.42 | 15.24 |
| 1168Ou-ASE | 13.45 | 46.13 | 4.06 | 3.36 | 5.75 | 77.58 | 1.27 | 44.91 | 2.83 | 0.45 | 16.84 |
| 1406Dv | 10.00 | 50.55 | 2.97 | 1.79 | 7.27 | 762.07 | 1.31 | 77.71 | 2.92 | 0.94 | 17.45 |
| 1406Dv-ASE | 10.17 | 47.56 | 3.00 | 1.79 | 22.81 | 984.84 | 1.32 | 89.88 | 4.27 | 1.27 | 24.40 |
| 926Men | 10.93 | 65.79 | 3.63 | 2.49 | 17.01 | 842.79 | 1.45 | 212.23 | 6.90 | 2.43 | 52.53 |
| 926Men-ASE | 10.78 | 63.99 | 3.65 | 2.57 | 8.23 | 825.24 | 1.47 | 197.98 | 6.10 | 2.42 | 53.06 |
| 926Tt | 11.20 | 79.94 | 4.27 | 3.27 | 13.23 | 462.44 | 1.14 | 105.72 | 1.34 | 1.65 | 31.74 |
| 926Tt-ASE | 10.36 | 77.49 | 4.22 | 3.15 | 7.54 | 421.90 | 1.14 | 93.07 | 1.36 | 1.47 | 30.73 |

Mg/Ca ratios were consistent for samples consisting of a large number of individual foraminifera, which averages out interspecimen variability. In samples with smaller numbers of individuals, specifically the *G. miotumida* and *O. universa* from core 1168A, there were resolvable offsets of 16.9 % and 27.4 % respectively. Mg is widely observed to be incorporated into foraminiferal laminar calcite in bands, the formation of which has been linked to diurnal processes in planktic foraminifera, including *O. universa* (Eggins et al., 2004; Spero et al., 2015). In contrast crust calcite, such as that exhibited by *D. venezuelana*, tends to lack high-Mg bands and therefore has a lower Mg-content relative to the laminar calcite (Eggins et al., 2003; Steinhardt et al., 2015). While there is an established and robust relationship between foraminiferal Mg/Ca and temperature (Nurnberg, 1995; Elderfield & Ganssen, 2000; Anand et al., 2003; Gray & Evans, 2019), it has been well documented that individual foraminifera that have grown in the same environment may record quite different bulk Mg/Ca values from each other (for example Weldeab et al, 2014; Davis et al., 2017; Davis et al., 2020). The impact of interspecimen variability on Mg/Ca was demonstrated by Rongstad et al. (2017) who performed single-foraminifera analyses of samples consisting of between 66-70 individuals across three species of foraminifera (*G. ruber, N. dutertrei, P. obliquiloculata*), on 9 samples in total. The spread in Mg/Ca values for individual foraminifera that they found for each sample ranged from 1.92 to 4.31 mmol/mol, with the biweight standard deviation (which reduces the effect of outliers) ranging from 0.37 to 0.83 mmol/mol. As this effect will be greater for elements which display a high degree of intratest variability, it will be muted in Mg/Ca measurements of species

with a relatively thick and homogenous crust of low Mg calcite. This may explain the close agreement within pairs for *D.*
*venezuelana* even when the number of specimens analysed is low, as is seen for samples from 1168A (pre-ASE n = 20, ASE
n = 28). Given that specimen numbers were extremely limited for *O. universa* (pre-ASE n = 13, ASE n = 16), it is probable
that in this case the offset in Mg/Ca values between the treatments is due to interspecimen heterogeneity, rather than any impact
of the ASE-treatment. The same process may contribute for the modest number of *G. miotumida* specimens. We note that due
to differences in trace element distribution within the test (e.g. Hathorne et al., 2009) and different environmental controls on
incorporation, there may not be a consistent offset among the trace element results for a given sample pair, or between species.

Sr/Ca ratios were elevated in ASE-treated *D. venezuelana* from 1168A (1.2 %) and *G. menardii* from 926B (1.5 %), but
depleted in ASE-treated *O. universa* from 1168A (1.0 %). U/Ca is elevated in ASE-treated *G. miotumida* (21.9 %), *O. universa*
(10.5 %) and slightly reduced in ASE-treated *D. venezuelana* from 1168A (8.4 %). The elevation of 39.8 % in ASE-treated *D.*
*venezuelana* from 1406B is less likely to be related to specimen number, but may be linked to elevated Mn/Ca in this sample,
discussed below.

Mn/Ca is offset is greater than uncertainty in all sample pairs (Fig. 5a), with the most striking offset being for *D. venezuelana*
from 1406B (ASE sample 29.2 % [222.77 μmol/mol] elevated). Regardless of treatment, all species pairs from 1406 (*D.*
*venezuelana*) and 926B (*G. menardii* and *T. trilobus*) showed elevated Mn/Ca values (> 400 μmol/mol), which are significantly
higher than generally accepted for foraminiferal samples unaffected by diagenesis (Fig. 5, Table 2). Additional authigenic
manganese may accumulate on foraminifera under changing redox conditions as either Mn-oxides or Mn-carbonates (Boyle,
1981; Boyle, 1983; Morse et al., 2007). Given the offset in Mn/Ca between *G. menardii* and *T. trilobus* from 926B, but
consistency within the pairs, it seems likely that Mn is hosted in these samples as authigenic Mn-carbonate, rather than Fe-Mn
oxyhydroxides, which would be unlikely to reproduce to the level shown (2.1 % for *G. menardii* and 8.8 % for *T. trilobus*).
This is supported by the presence of pink crystals on the exterior of some specimens, which were noted in individuals from
both untreated and ASE-treated samples (Fig. 1), and the absence of internal features such as pores visible in the SEM image
of ASE-treated *T. trilobus* (Fig. 2d). Together these might indicate the precipitation of one or more secondary mineral phases.
The offset between the measured Mn/Ca of the species is likely due to differences in Mn/Ca of the host phase caused by either
differing morphology or geochemistry (organics or trace elements) of the specimens at deposition. However, Fe-Mn
oxyhydroxides may be the source of high Mn in the samples from 1406B based on the large offset between the samples and
the lack of a visible authigenic phase identified using SEM; fine scale contamination with discrete Fe-Mn oxyhydroxides might
not be detected using SEM imaging. Although authigenic phases such as Fe-Mn oxyhydroxides may also be enriched in other
trace elements including Mg (Pena et al., 2008; Roberts et al., 2012), we observe no relationship between Mn/Ca and Mg/Ca.
Elevated Mn/Ca is not reflected in greater values for Mg/Ca, and Mg/Ca remains within the range expected for a primary
signal (Fig 5b; pre-ASE $R^2 = 0.04$, ASE $R^2 = 0.03$).

The exact cause for the elevated Mn/Ca is unclear. The sulfate-methane transition zone (SMTZ) can influence the precipitation
of secondary authigenic phases, but seems unlikely to have done so at these sites. The SMTZ is found at 225-230 metres below
sea-level (mbsf) at ODP Site 1168A, meaning that *D. venezuelana* from 1168A-25X4 50-52 lie around the SMTZ and *G.*
*menardii* and *O. universa* from 1168A-26X4 50-52 slightly below it (Shipboard Scientific Party, 2001). However, these
samples showed no visible signs of authigenic phases nor had significantly elevated Mn/Ca to indicate this. There is no analysis
of methane or sulfur at site 1406B, only for 1406A, but based on stratigraphic correlations between both sites, the SMTZ
would be at around 161 m (approximately) in 1406A, and in an equivalent depth at 1406B (Norris et al., 2014). Therefore our
sampled depth sits above the SMTZ. For 926B sulfate concentrations decrease by nearly 70% over the sampled sequence at
926B (down to 591.25 mbsf), but sulfate is never fully reduced (Shipboard Scientific Party, 1995). There therefore appears to
be no relationship between the location of the SMTZ and the samples with elevated Mn.

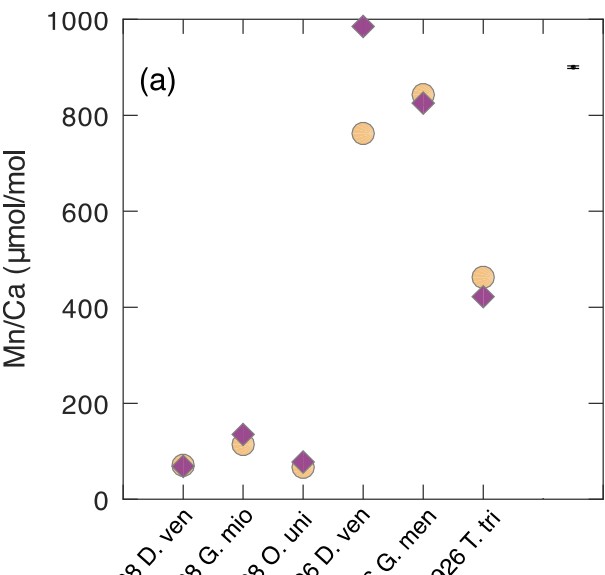 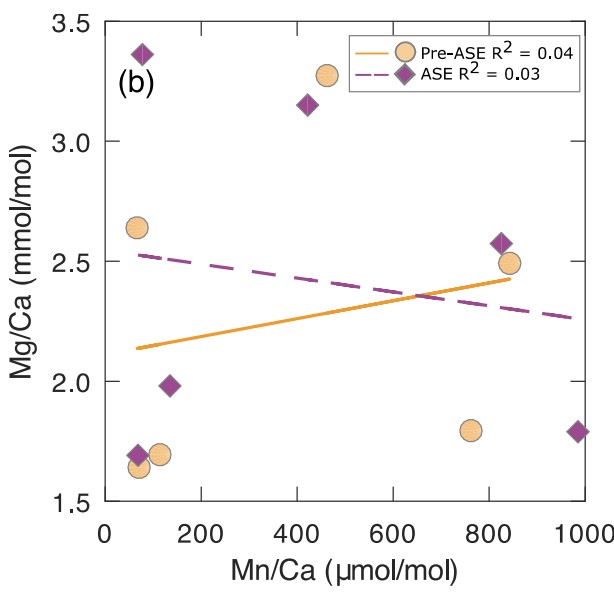

**Fig 5. (a) Mn/Ca results for the pre-ASE (yellow circles) and ASE-treated (purple diamonds) foraminifera. (b) There is no discernible**
**relationship between Mn/Ca and Mg/Ca, indicating that despite high Mn/Ca values indicative of secondary authigenic signals,**
**foraminiferal Mg/Ca is not affected. Solid yellow line = pre-ASE linear fit, dashed purple line = ASE linear fit.**

### 3.4 Boron isotope results

We find that no sample pairs exceed 2SD difference between the treatments for $\delta^{11}B$ (Fig. 6, Table 2). In all cases, non-treated
and ASE $\delta^{11}B$ values fall within 2SD of each other. Note that the larger uncertainties for *O. universa* and *G. miotumida* are
due to the small mass of these samples. We note that there is no discernible impact of the ASE treatment on either the boron
concentration (Fig. 4b) or boron isotopic composition (Fig 6.), which is consistent with the incorporation of boron into the
carbonate lattice (e.g. Branson et al., 2015) allowing for the preservation of the signal despite the high temperatures
experienced during ASE treatment. Minor offsets in $\delta^{11}$B between different species of planktic foraminifera from the same
interval (e.g. between different species in the samples from 1168A and 926B) are expected and reflect the combined influence
of differences in depth habitat in the water column, and the presence or absence of photosymbionts (Henehan et al., 2016).
The positive offsets seen in the trace element measurements of ASE-treated *G. miotumida*, *O. universa* and 1406B *D.*
*venezuelana* are not observed in the boron isotope data. The boron isotope measurements here show no evidence that ASE-
treatment impacts foraminiferal $\delta^{11}$B and indicate that reconstructions of ocean pH made from ASE-treated foraminifera can
be applied with confidence.

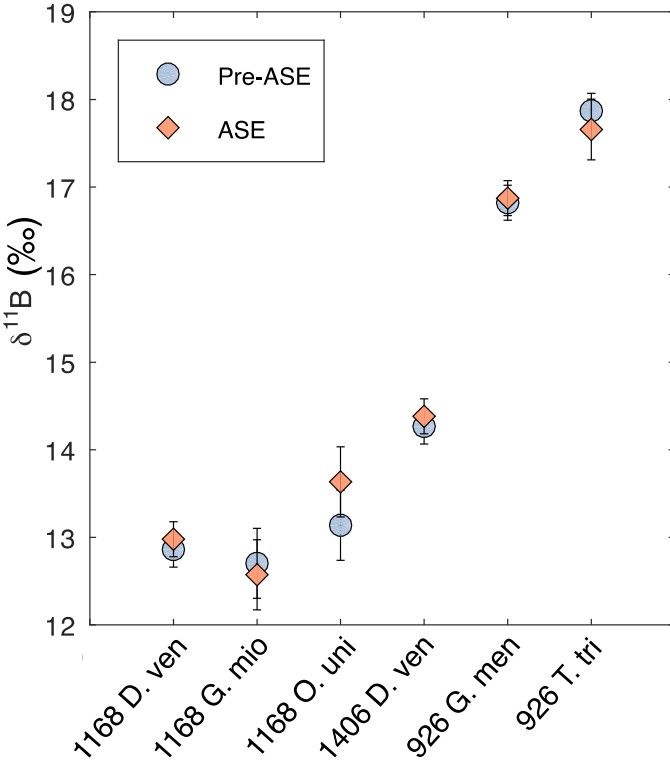



**Fig 6. Boron isotopes results for comparison of pre-ASE (blue circles) and ASE treated (pink diamonds) foraminifera. No offset**
**between the two treatments is identified, supporting the use of foraminifera from ASE treated sediments for boron isotope**
**reconstructions of pH.**

## 4 Conclusions

We have undertaken a detailed assessment of the potential impact that ASE treatment of sediments has on the geochemistry and physical preservation of the planktic foraminifera hosted within that sediment. We see no signs that ASE treatment leaches or dissolves foraminiferal calcite and find no evidence that it discernibly influences geochemistry. Some ASE-treated samples do exhibit slightly elevated trace element values, but this is likely due to a small number of individuals analysed. Our findings support the use of foraminifera from ASE-treated sediments for geochemical analyses, including Mg/Ca to reconstruct temperatures, and $\delta^{11}B$ to reconstruct pH. These findings pave the way for paired Mg/Ca and GDGT or alkenone derived temperatures, and $\delta^{11}B$ and alkenone $\varepsilon_p$ reconstructions of the carbonate system from the same sediment. We hope that this study will give the scientific community confidence to share ODP samples that might otherwise be discarded as waste and pave the way for new collaborative endeavours.

**Data availability:** All data from this study is published in the tables within this manuscript.

**Author contributions:** JCB conducted the SEM preservation study, trace element and boron isotope analyses, interpreted those data, and wrote the first draft of the manuscript. TT conceived and designed the study, procured samples, completed lipid extractions and obtained and interpreted stable isotope analyses. IHA identified and picked foraminifera. All authors contributed to the final manuscript.

The authors declare that they have no conflict of interest.

**Acknowledgements**

This research was funded by the Swiss National Science Foundation (Award 200021_182070 to HMS) and ETH core funding. JWBR received funding from the European Research Council under the European Union's Horizon 2020 research and innovation program (grant agreement 805246). We acknowledge the support of the EPSRC Light Element Analysis Facility Grant EP/T019298/1 and the EPSRC Strategic Equipment Resource Grant EP/R023751/1, which supported SEM analyses at the University of St Andrews.

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
