# Peer review of "Technical Note: No impact of alkenone extraction on foraminiferal stable isotope, trace element and boron isotope geochemistry"

_Biogeosciences, 2022_

## Author Response (AR1)

Reviews – summary and response. Line numbers refer to the tracked changes document.

**Reviewer 1**

1. *Page 1, lines 24-26: the logic of the grouping of proxies between clauses a) and c) seems a bit unclear. Also what about micropalaeontological/assemblage work, like transfer functions etc.?*

   Our intention here was to introduce the proxies that we worked with in this study, and so those are the only proxies we discussed in this section. We have amended this sentence to improve the clarity of our intention (Lines 23-29]

2. *Section 2- the first sentence of this detailing where all the samples were taken is far too long and unwieldy, with too many clauses and brackets. Suggest breaking up for the benefit of the reader.*

   We agree and have revised this description for clarity [Lines 51-59]

3. *Page 3, Line 71: Can the authors please clarify if these growths were seen only in the ASE-treated samples, or in both the ASE and non-ASE samples?*

   Thank you for pointing this out, we mention this in the caption and later in the discussion but agree that it should be mentioned here as well, and have amended this sentence to clarify this [Lines 116-119]

4. *Page 5, Line 109-110: the authors mention here the Al/Ca measurement results, and then mention it again later on Page 6, Lines 130-131. Could the authors perhaps avoid repetition by saying 'despite the shorter ultrasonication time in some samples' at the end of the Page 6 line 130-131 instead?*

   Thank you for this suggestion, we have included it in the revised manuscript [Lines 171-172].

5. *Page 6, Line 128: Please give the absolute values of these measured El/Ca ratios as well as the % variability between measurements.*

   We have incorporated these into our revised manuscript [Lines 166-168].

6. *Table 2: Why should d18O and d13C be reported 1sigma, and boron 2? I have never understood why O/C isotopes should be held to a lower standard.*

   We are happy to amend this and have reported the oxygen and carbon isotope results to 2 sigma [Table 2; Line 135].

7. *Table 12, Line 260: give reference for this statement – e.g. https://www.nature.com/articles/ncomms15441?*

We agree that this statement should be referenced, and we have included additional citations in the revised manuscript [Lines 318-319].

8. *Page 1, Line 45: foraminiferal geochemistry.*

Thank you for pointing out this error, we have corrected it in the revised manuscript [Line 49].

**Reviewer 2**
9. *Line 50: be consistent with naming of sample ID, compared to line 48.*
10. *Line 52: same as above*

Thank you for highlighting this, we have amended this in our revised version [Lines 51-59].

11. *Line 54: you state here that the samples were washed with miliQ and in line 59 you used deionized water.*

Apologies for the confusion; all sieving was carried out using deionised water and we have amended this in our revised version [Lines 58-59].

12. *Lines 56-59: I would like to see a better description for the extraction method: How long was each static cycle? What was the amount of solvent in the cell? What cell size? Would you expect it to also show no effect on different cell sizes, amount of sediment and number of static cycles?*

Incorporating these and comments from the other reviewers and editor, we have updated and amended section 2.1 to the following [Lines 61-84]:

2.1 Solvent extraction of lipids

The sediment was gently disaggregated while still in the plastic bag from the repository before freeze-drying and is not exposed to any other plastic or glass during the whole pre-sieving process. After freeze-drying, the sediment, still inside the plastic bag, was crushed into small grains using a small rubber mallet to homogenize the sample and increase the surface area for extraction. It is not further grinded down into a fine powder to preserve the various microfossils. Clearly, there is a tradeoff between the efficiency of alkenone extraction and the preservation of intact microfossils. The method described here does not involve extreme grinding because it tries to avoid destruction of microfossils and has been used successfully in various publications (e.g. Guitián et al. 2019, Tanner et al. 2020, Guitián et al. 2021). This procedure may reduce the exposure of alkenones within the sediment to the organic solvent and hence reduce the extraction efficiency, compared to extreme grinding with pestle and mortar. However, any variations in extraction efficiency would not change the result of paleoceanographic proxies that are based on the ratios of organic compounds.

Half of the freeze-dried sediment with a dry weight between 18g to 24g, was extracted using a Thermo Dionex 350 accelerated solvent extractor at the Department of Earth Sciences of ETH Zürich. Therefore, the sediment was put in 34ml stainless steel cells and extracted with three 10-minute static cycles at 100°C with a 5:1 ratio of dichloromethane to methanol (DCM/MeOH). The DCM is a biotech grade solvent (602-004-00-3) from Honeywell and the MeOH is a liquid chromatography grade solvent (1.06007.2500) from Merck KGaA. After three cycles, each extraction delivered a total solvent volume between 85ml to 90ml. We are confident that after three cycles, most of the organic material is extracted from these carbonate rich sediments. Working with similar sediment showed that ~ 90% is extracted with the first cycle and that the second and third cycle extract the remaining ~ 10%.Similar results have been reported by Auderset et al. 2020. Subsequently, the now treated sediment was sieved with deionized water through a 150 μm sieve and oven dried overnight at 50°C. The target organic compounds, such as alkenones, are not contaminated by any plastics from the repository bags, because plastic derivatives have much shorter retention times and would elute much earlier in a gas chromatography column. Although alkenones have not been measured in this study, in several years of using this procedure (e.g. Guitián et al. 2019, Tanner et al. 2020, Guitián et al. 2021), plastic contamination was never observed in the earlier part of the chromatograms during alkenone analysis.

13. *Table 1:*

    *I don't understand the "number weighted and analysed" term. Could you explain that a bit better in the text or choose another title in the table?*

    The picked foraminifera samples were initially counted and weighed, but subsequent SEM analysis and ultrasonic tests reduced the number to be analysed; for *T. trilobus* and *G. miotumida* additional specimens were also picked, so the final numbers of these that were analysed increased from those that were weighed. The final sample of foraminifera were not reweighed prior to analysis. Because we interpret our results as having been influenced by the number of individuals analysed, it seemed important to us to give this number accurately, which was counted from light microscope images that were taken just prior to analysis. We have amended the text in section 2.2. to clarify this process [107-111]

14. *Be consistent with the naming of the core, also add the Leg number to Site 926 and 1406. Add. Hole to 1406. Should be hole B?*

    Thank you for drawing our attention to this, we have edited the names within the table to be consistent [Table 1].

15. *Be consistent with the succession of the cores in the Tables and Figures. It makes it easier for the reader to look up values between tables and figures. Change the succession for Table 1, starting with 1168 ->1406 -> 926, like you do for the other tables and figures.*

Thank you for highlighting this, we have amended the table to be consistent [Table 1].

16. *Fig. 1: are T.trilobus and G.menardii in a) and b) treated or untreated? And could you add this information to the caption?*

    Both of the light microscope images are of foraminifera from untreated sediment. We have clarified this within the caption [Lines 116-119].

17. *Line 200: If the d13C and d18O variability is attributed to small sample size, wouldn't we expect the d18O for O.univ also to be high?*

    Not necessarily; this depends on the variability within the populations for $d^{13}C$ and $d^{18}O$, which will not necessarily be the same for each parameter. The offset in $d^{13}C$ might be driven by one outlier sample in which $d^{18}O$ is not elevated. The impact of such an outlier would be greater in a smaller population, such as we have with *O. universa*.

18. *Line 204: (Fig. 4b)*

    We are unsure what the reviewer is referring to here, but guess that they would like us to reference figure 3b here, which we are happy to do [Line 249].

19. *Lines 230-271: it would help the reader to follow the figure in chronological order. For example, in Section 3.3 the authors describe Fig.4b->e->c->a->d->f. Wouldn't it be easier to sort the figures in the order they are discussed or is there a reason for the sorting now?*

    The subpanels shown in the figure are sorted by atomic mass, and in text are discussed loosely based on similarities to each other, however we have amended this to match the sequence given in the panels [Lines 300-340].

20. *Lines 230-271: Can the authors doublecheck the difference values (in percentage) in this section? In some cases I don't get the same values when calculating them from the results in Table 3.*

    We thank the reviewer for pointing this out. While reviewing this section, we identified an error where all the offset values have been reported relative to the untreated samples; when the offset is stated as an ASE increase, the percentage value therefore appears incorrect. Furthermore, the values for uncertainty on the individual Sr measurements were reported in decimal notation, rather than as a percentage. We have amended these points in the revised version. We have also reported all percentages to 1 decimal place. Please let us know if these changes do not address your concerns [throughout Lines 300 – 340].

21. *Line 245 (Fig. 4c)*

    *Line 280: (Fig. 2d)*

We have included reference to the subpanels in the text where they are referred to [Line 307, 352].

22. *Line 312: A nice addition to the conclusion would be to point out that you can not only compare alkenone ep with $\delta^{11}B$, but also Mg/Ca temperatures with GDGT-derived and alkenone-derived temperatures as the authors mentioned in the introduction.*

We thank the reviewer for this suggestion, and have incorporated it into our conclusion [Lines 403-404].

23. *General comment:*

*I am not fully convinced of the explanation that the sample size caused the differences in G.mio and O.uni in 1168. For Mn/Ca D.ven seems to have a big offset despite its big sample size. For Li/Ca and U/Ca O.uni seem to have no significant offset despite their small sample size. But due to the lack of consistent offset between treated and untreated sediments I agree that the ASE treatment should have no significant effect on the trace metals, especially not on the boron isotopes. A more in-depth study of the samples with bigger offsets would be interesting for the future.*

Some elements (such as Mg/Ca) are distributed more heterogeneously in foraminifera. There may also be varying degrees of influence by multiple environmental drivers, resulting in disparate expression in trace elements [Lines 332-334].

Addressing the point about Mn/Ca in D Venezuelana from 1406B: the lack of obvious secondary precipitation as in 926B, and the offset between the Mn/Ca values, suggests that the Mn in samples from 1406B might be hosted in an Fe-Mn oxide rather than a carbonate, which is what we suggest for 926B. We have amended our discussion to include this [Lines 354-357].

For the comment on Li/Ca and U/Ca: similarly to our response to the comment on stable isotope offsets above, we would like to stress that a smaller sample size acts to increase the impact of interspecimen variability on the resulting data. The degree of heterogeneity between one foraminifera and the next is likely to be greater for elements which display larger intratest variability, such as Mg/Ca.

Li/Ca appears to exhibit less intratest variability within some planktic foraminifera species than Mg/Ca (e.g. Hathorne et al., 2009), suggesting that there is likely to be less Li/Ca variability between individual foraminifera than we might expect for Mg/Ca. We suspect something similar for U/Ca.

We have elaborated on these points within the revised manuscript, and hope that this addresses your concerns.

24. *Do you change the resultant measured alkenone unsaturation index (or GDGT ratio, or isotopic composition, or any other lipid/biomarker value) by not crushing? You could, for example, not get complete extraction and one alkenone is preferentially extracted over the other?*

*Similarly do you reduce the yield of alkenones by not crushing? This would both reduce the utility of the method (as you want to get as much as possible), but also make it more difficult for anyone wanting alkenone concentration, as the extraction efficiency will likely be heterogenous down core if you don't homogenise.*

*It is possible that these questions have been answered elsewhere – that someone has experimented with crushing vs not crushing for ASE alkenone/lipid extraction and it matters not a bit, but if that is the case those experiments need citing here, and if it hasn't been tested, at the very least a discussion of the above would be helpful.*

*It's a pity, because the logical addition in this study would have been to crush an aliquot of the sediment, extract in the ASE, and then compare the alkenone data between the crushed and not-crushed samples. This could have reassured both the organic and inorganic geochemists in the authors proposed new collaborative endeavours.*

*Nonetheless, this is a neat study and should be published once these minor concerns have been dealt with.*

These are all valid concerns and we thank the reviewer to bring these forward. It is in fact true that grinding the sediment would result in a fine powder with most of the foraminifera destroyed. That is why we only use a small rubber mallet to crush the sediment into small pieces that still contain intact foraminifera. It is obviously a trade-off between preserving the microfossils and extracting as much organic material as possible.

Also, this seems to be standard practice in the lab. The cited Zhang et al. 2017 paper for example describes using a mortar and pestle to homogenize the sediment (but not making a powder) and later using the >250ym size fraction to pick planktonic foraminifera.

It is highly unlikely that the alkenone ratio would be affected by imperfect extraction. The various lipids behave very similarly and all of them are easily dissolved and extracted using the described 5:1 DCM:MeOH ratio. Even though the extraction might be imperfect by not crushing the sediment entirely, we are showing that we can compensate for it by using much more sediment to extract from because it is not affected by the solvents.

We have amended the methods section to address these concerns [Lines 61 - 84], and hope this is satisfactory.

*Minor points*

25. *Section 2.1 This section is rather short, especially compared to the detail in the boron methods. What was the mass of the samples? What volume of solvent was used? What volume of ASE inserts were used? What grade or solvents were used and who supplied them?*

    Thank you for highlighting your concerns about this section. We have amended the section to address these points [Lines 61 - 84].

26. *Line 160 I wonder whether some of this apparent better preservation amongst the ASE treated samples is because you've plausibly done a light organic matter clean. Plausibly this could effect either their appearance, or how well they will sputter coat?*

    This is an interesting alternative explanation. We have taken pains to indicate that the effect we noticed was slight, and with the caveat that the sample sizes were small (and the effect may therefore be due to the small number analysed). We note though that for *T. trilobus* the gaping observed between the layers appeared slightly greater in the pre-ASE versus ASE samples, which is the opposite to what we would expect to observe if the loss of organic layers was making a visible impact (or causing the difference in preservation). However, the difficulty in identifying features in the pre-ASE samples might be influenced by the presence of organics and debris on the faces, and we have mentioned this in the revised manuscript [Lines 237 – 240].

27. *Lines 193-8: This result is surprising. At this point I would have wished I had done SEM on more than 3 individuals? I presume by the time this realisation was made the rest had been thoroughly dissolved?*

    As the geochemical data shows, this slight feature is not important for the conclusions of the paper, and a detailed SEM study of it is outside the scope of this study. This could be an interesting avenue for future studies, with steps taken to minimise contamination by fragments of the surface faces.

28. *Figures (generally). A crossplot of Pre-ASE vs ASE would be informative, potentially with some stats too. I've done a quick example of the data in Table 2 in the plot below. A similar plot for the Table 3 data might be more informative than Fig. 4. Incidentally would Not-ASE vs ASE be clearer terminology?*

    We appreciate this suggestion from the reviewer, although our initial concern with displaying the data in this way was that it is less easy to ensure that the identity of the species/core treatment pairs is clear and accessible to all readers. This is due to the large number of unique markers required (six) and overlap between the markers in some instances leading to markers being obscured. We also note that Reviewer 1 states the paper is "clearly presented" and Reviewer 2 that "The figures are clear".

Respectfully, we disagree that not-ASE would be clearer terminology than pre-ASE. We feel that the usage of pre- is not misleading as we are referring to samples before and after the ASE treatment process.

29. *Line 291 "We find no significant difference between the treatments". You have not done a significance test so you should not state this. You should do a significance test and then you probably can!*

We will amend this sentence to "We find that no sample pairs exceed 2SD difference between the treatments for $d^{11}B$" and hope that this addresses your concerns [Line 379].

**Reviewer 4**

30. *The reproducibility of trace elemental ratios presented in this study for six individual sample sets between ASE-treated and non-ASE treated sample sets is remarkable, but the authors somewhat oversell the reproducibility for Li/Ca and Mg/Ca. For both elemental ratios, for two out of six (i.e., one third) of the samples the reproducibility is in fact not good. The authors argue for smaller sample sizes (256-257, 265-267) but this is not apparent from the plots. As long as no other geochemical criterion can be identified to distinguish a reliable from an unreliable elemental ratio for ASE-treated samples such elemental ratios from ASE-treated samples should not be used in such a manner, and the authors should make this clear in the manuscript. The other ratios are either remarkably (!) good (B/Ca, Na/Ca, Sr/Ca) or satisfactory (U/Ca).*

We appreciate the reviewer sharing their thoughts on this. A large range in Mg/Ca variability between foraminifera grown under the same conditions is well documented, for one example see the ~1 mmol/mol spread in values of samples of 20-30 *G. ruber* individuals at ~28 C from plankton tows in the Mozambique Channel (Weldeab et al, 2014).

Spread such as this are likely driven by large interspecimen variability in Mg/Ca. Such variability was demonstrated by Rongstad et al. (2017), who performed single-foraminifera analyses of samples consisting of between 66-70 individuals across three species of foram (*G. ruber, N. dutertrei, P. obliquiloculata*), 9 samples in total. The spread in Mg/Ca values for individual foraminifera that they found for each sample ranged from 1.92 to 4.31 mmol/mol, with the biweight standard deviation (which reduces the effect of outliers) ranging from 0.37 to 0.83. Given this, it makes sense that we see an offset in Mg/Ca between our samples of *O. universa*, numbering 13 and 16 individuals.

We also note that for Li/Ca, there is no consistency in terms of the direction of the offset. Of the two samples which have an offset greater than the uncertainty on the measurement, one is observed in an ASE samples, and one in a pre-ASE sample.

Feedback from a previous reviewer is that the order of samples within the tables and plots should follow a consistent sequence, and for the ease of the reader, we agree with this approach. However, we suggest that we indicate on the plot which of the sample pairs includes a sample which consists of <70 individuals, and hope that this will be acceptable to the reviewer. We have expanded on these points about interspecimen heterogeneity and the impact when analysing small samples in the revised manuscript [319-3328].

31. *No reductive cleaning was carried out. While I understand why the authors did not carry out this step (avoidance of signal bias for Mg/Ca as well as sample loss) this renders the Mn/Ca signal at the very least ambivalent if not pointless. The authors discuss the features of the Mn/Ca results but should not place too much weight on recovered ratios. It is an interesting observation that ASE treatment did not affect the sample Mn/Ca for five out of six sample sets, but such a ratio should probably not be interpreted in a palaeoceanographic fashion anyways since the largest proportion of the Mn in the signal is likely derived from Fe-Mn oxyhydroxides attached to the foraminifera. I also have reservations towards the argument that Fe-Mn oxide phases are not detectable in SEM images. These may simply be too fine-scaled (284-285) to be detected with SEM imaging. If a sediment contains Fe-Mn oxyhydroxides as well as authigenic Mg-containing mineral phases I would not expect a correlation (285) since these are two separate properties of the sediment with independent origin.*

We thank the reviewer for raising this important point, and agree that this section would benefit from rewording. We lay out our main points here, and trust that incorporating these into this section will address the reviewer's concerns.

We agree that as a palaeo-signal, the Mn/Ca is not of palaeo-proxy interest given the lack of a reductive clean. We also tried to avoid placing too much weight on these ratios given the presence of the unusual pink crystalline growths shown in Figure 1 which are potentially linked to the elevated Mn/Ca in samples from core 926B. We appreciate that we failed to make the (potential) link between the pink crystals/authigenic coating and elevated Mn/Ca, which we have amended in our revised manuscript.

Our main argument for a carbonate phase hosting Mn rather than Fe-Mn oxyhydroxides in the samples from core 926B is the remarkable agreement between the Mn measurements (coupled with the authigenic coating observed under SEM, and visible crystalline growths), and the fact that there is this agreement within the species pairs but not between them. We believe that this is unlikely to occur in these measurements without a constant ratio of Mn:Ca, such as would be found in a carbonate, and would be unlikely in the case of Fe-Mn oxyhydroxides. It is plausible that different geometry or starting geochemistry (organics or trace elements) in the foram species might explain the consistent offset between *G. menardii*and *T. trilobus* from 926B.

We do however suspect Fe-Mn oxyhydroxides may be the source of high Mn in the samples from 1406B, based on i) the elevated Mn/Ca, ii) the offset between these

samples and iii) the lack of any visible contamination either under light microscope or SEM; as the reviewer states, these may not be detectable using these tools.

The reviewer raises an important point regarding our interpretation of the Mg/Ca:Mn/Ca crossplot, and we thank them for bringing this to our attention. As we have assumed a single secondary phase, we had not considered that there might be issues with exploring this using a crossplot. However, we do feel that there is utility in the cross-plot; this shows clearly that while Mn might be significantly elevated, Mg/Ca remains reproducible within the species pairs and within the range we would expect for a primary signal, though differs between species (as would be expected by the occupation of different ecological niches; an investigation into these was considered but we decided is outside the scope of this paper). We have amended this section to include the points we have outlined above [Lines 348 – 360].

32. *The reproducibility of the boron isotopic results is outstanding and a key result of this study. On the other hand, the discussion around it is really short. Is there really nothing to discuss? The authors should at the very minimum also include a d11B vs B/Ca plot. They should also mention that the triple treatment of the ASE samples indeed had no (coupled) isotopic or B/Ca effect. This is not obvious from a theoretical point of view given that boron is volatile and the likelihood of a fraction of boron escaping the carbonate matrix during this pre-treatment is not zero and could hence be mentioned.*

We thank the reviewer for their interest and appreciation of the reproducibility in the boron isotope results. It is indeed an interesting point that the lack of an ASE impact suggests this treatment really does not release any boron from the carbonate matrix to be volatilised, though this is consistent with the apparent good preservation state of the foram shells following treatment and boron's incorporation into the CaCO3 lattice (e.g. Branson et al., 2015, EPSL). We have included a brief note on this in the $\delta^{11}B$ results [381-384]. We would be happy to include an additional plot and further discussion, but are concerned about overly expanding the length of the manuscript given it was submitted as a Technical Note and so leave this to the Editor's discretion.

*Minor comments:*

33. *Lines 24-27: Why not add two good example references behind the usage of every proxy/parameter?*

We thank the reviewer for this suggestion, and have included these [Lines 25-29].

34. *Line 33: "the 10 Myr residence time of boron in seawater presents a challenge for determining absolute ocean pH values on multi-millennial timescales" – add brief explanation as to why this is the case. It is not immediately evident to every reader.*

We have added a brief statement of elaboration (" as changes in foram $d^{11}B$ will be a function of both changes in pH and in the $d^{11}B$ of seawater"). We also reference

papers that describe this effect in detail for those who are interested to pursue the details further. We thank the reviewer for drawing our attention to this sentence, as we have noticed an error: this should read multi-million year timescales [Lines 36-37].

35. *Line 34-35: "while phytoplankton-based proxies may struggle to capture low-CO2 conditions" - add brief explanation as to why this is the case. It is not immediately evident to every reader.*

We have added a further brief explanation to this point, and have referred the reader to further sources of interest [Lines 38-39].

36. *Line 58: CH2Cl2/MeOH sounds like a mixture of a molecular formula and an ingredient name. The molecular formula would be C2H6Cl2O, the name one of these: DCM methanol, methanol DCM, CH2Cl2 methanol, methanol CH2Cl2, dichloromethane MeOH. So how would this reagent officially be addressed?*

We thank the reviewer for pointing this out, and propose to amend this to DCM/MeOH [Line 74].

37. *Also line 58: I find it remarkable that triple sample treatment at 100°C (how long actually?) does not affect the boron isotopic composition of the carbonates. This needs mentioning in the discussion (see major point above).*

Each static cycle took 10 minutes, so a total of 30 minutes was spent at 100° C. We note that no such impact has been noted during oxidative cleaning which typically occurs at rather elevated temperatures itself (here, 80 °C for 15 minutes) . As mentioned above, the lack of boron release at these temperatures is consistent with its lattice bound position [see Lines 74; 381-384].

38. *Table 1: Please also add the depth downcore of each sample used. Furthermore, it would be useful to know whether these sedimentary depths were positioned below a possible sulphate-methane transition zone (if present at these sites). I am mentioning this since sediments within the methane stability field may contain authigenic carbonates (e.g., Meister et al., 2007, Sedimentology) which could have an effect on the nature and robustness of an extracted stable isotope (B, C or O) or trace element signal. The authors for example mention the possible presence of an authigenic phase in lines 165-167. Could the presence of authigenic carbonates for example have consequences for some of the non-reproducible Mg/Ca or Li/Ca in the sample set?*

Thank you for raising this point. Based on this and feedback from a previous reviewer, we have amended the sample table to give the Leg, site, hole, core, section, interval, and depth of the cores. The depth of the samples is summarised here:

1168A 25X = 229.2 m

1168A 26X = 238.8 m

1406B = 63.04 m

926B = 185.41 m

The details of the sulfate-methane transition zone (SMTZ) of the cores used here are as follows. The SMTZ it is found at around 225-230 mbsf at ODP Site 1168A (http://www-odp.tamu.edu/publications/189_IR/chap_03/chap_03.html), so 25X would be at around the SMTZ, and 26X immediately below it. However, the 3 samples from this site had the lowest Mn analysed, and no visibly identified evidence of authigenic carbonates.

For 1406B there is no analysis of methane or sulfate for that site, only for 1406A, but based on stratigraphic correlations between both sites, the SMTZ would be at around 161 m (approx) in 1406A, and in an equivalent depth at 1406B (http://publications.iodp.org/proceedings/342/107/107_f26.htm). Therefore our sampled depth sits above the SMTZ.

For 926B sulfate concentrations decrease by nearly 70% over the sampled sequence at 926B (down to 591.25 mbsf), but sulfate is never fully reduced (http://www-odp.tamu.edu/publications/154_IR/VOLUME/CHAPTERS/ir154_05.pdf)

There appears to be no relationship between the location of the SMTZ and the samples with elevated Mn. It could potentially be a factor in the offsets observed in 1168, given the proximity of these samples to the SMTZ, although we do not observe any indication of an authigenic phase in these samples, and believe the offsets here are much more likely to be due to interspecimen heterogeneity, as laid out above.

We have included a reference to the SMTZ in the text [Lines 362-371], although we feel that a detailed investigation into the cause of the authigenic phases lies outside the scope of this paper.

39. *Table 1: I also do not understand why for some samples more individuals have been analysed than were apparently weighed in. This does not make sense to me given the explanation in the text.*

In the text it is mentioned that further specimens were picked for *T. trilobus* and *G. miotumida*; however, we thank the reviewer for drawing our attention to the fact that this point could be made more clearly, and have amended the manuscript to do so [Lines 107-111].

40. *Fig. 1: Very interesting figure and feature of these dark crystal overgrowths.*

We thank the reviewer for their interest in this point; we agree that this is an intriguing feature and hope that the publication of this paper might stimulate interest in them.

41. *Lines 107/108: Five seconds is a very short exposure time of foraminifera shells to ultrasonication during cleaning! But the Al/Ca measured on cleaned foraminifera sound encouraging.*

We agree that this is a short period of time, and are also encouraged by the low Al/Ca.

42. *Line 123: better write "triple quadrupole" than "QQQ".*

Thank you for pointing this out, we have amended it in our revised manuscript [Line 161].

43. *Lines 127-129: Do these reproducibilities represent 1 SD or 2 SD?*

We thank the reviewer for drawing our attention to this, the reproducibilities here are 2SD. We have edited our revised manuscript to include this [Line 166].

Regarding the uncertainty on Mg, we have noticed an error in the reporting of this. We typically collect data for 24Mg and 25Mg as standard, and report the data for 24Mg, given the greater number of interferences on 25Mg, and that 24Mg tends to be more robust to switches in detector mode, and more stable long term as a result. However, we have noticed that in this case we had erroneously reported the error on 25Mg (0.89 %) instead of 24Mg (1.97 %), and we have amended this in the revised manuscript [Line 176]. The differences between the collected datasets are very small (<0.03 mmol/mol) and this will have no impact on the findings of our paper, but we wanted to ensure that this mistake is noted and corrected at this stage.

44. *Line 135: Please add molarity of ammonium acetate buffer. What was ammonium acetate buffered with at which concentration?*

Boron isotope column chemistry depends on adjusting the pH of the sample, which is dissolved in 0.5 M $HNO_3$. Amberlite resin sorbs borate ion, and releases boric acid; therefore, to capture the sample in the column, manipulation of the sample chemistry is necessary. This is achieved here by buffering the sample with a pH 6, 1.1M ammonium hydroxide:1.2 M acetic acid buffer (exact concentrations adjusted to achieve pH ~6). We have amended our methods to include this information [Lines 176-177].

45. *Line 143: I am surprised and impressed that the authors still managed to obtain a decent isotopic signal (after blank correction) from such a low-concentration boron solution.*

We thank the reviewer for recognising the quality of the data obtained here. We have a manuscript in preparation that details the methodological developments that have permitted reproducibility at this level.

**Editors comments**

46. We thank the editor for suggesting that we give further clarification of the organic extraction and propose to change section 2.1 as proposed below (new text in red, Lines 61-84). In further details, we can clarify that we don't see any channels in the sediment after extraction. We also have not observed blockage of sediment within the ASE-cells. Additionally, we are not using diatomaceous earth during the extraction process. We have included some additional references to highlight that the here described procedure is standard and has been used many times in the past.

2.1 Solvent extraction of lipids

The sediment was gently disaggregated while still in the plastic bag from the repository before freeze-drying and is not exposed to any other plastic or glass during the whole pre-sieving process. After freeze-drying, the sediment, still inside the plastic bag, was crushed into small grains using a small rubber mallet to homogenize the sample and increase the surface area for extraction. It is not further grinded down into a fine powder to preserve the various microfossils. Clearly, there is a tradeoff between the efficiency of alkenone extraction and the preservation of intact microfossils. The method described here does not involve extreme grinding because it tries to avoid destruction of microfossils and has been used successfully in various publications (e.g. Guitián et al. 2019, Tanner et al. 2020, Guitián et al. 2021). This procedure may reduce the exposure of alkenones within the sediment to the organic solvent and hence reduce the extraction efficiency, compared to extreme grinding with pestle and mortar. However, any variations in extraction efficiency would not change the result of paleoceanographic proxies that are based on the ratios of organic compounds.

Half of the freeze-dried sediment with a dry weight between 18g to 24g, was extracted using a Thermo Dionex 350 accelerated solvent extractor at the Department of Earth Sciences of ETH Zürich. Therefore, the sediment was put in 34ml stainless steel cells and extracted with three 10-minute static cycles at 100°C with a 5:1 ratio of dichloromethane to methanol (DCM/MeOH). The DCM is a biotech grade solvent (602-004-00-3) from Honeywell and the MeOH is a liquid chromatography grade solvent (1.06007.2500) from Merck KGaA. After three cycles, each extraction delivered a total solvent volume between 85ml to 90ml. We are confident that after three cycles, most of the organic material is extracted from these carbonate rich sediments. Working with similar sediment showed that ~ 90% is extracted with the first cycle and that the second and third cycle extract the remaining ~ 10%.Similar results have been reported by Auderset et al. 2020. Subsequently, the now treated sediment was sieved with deionized water through a 150 µm sieve and oven dried overnight at 50°C. The target organic compounds, such as alkenones, are not contaminated by any plastics from the repository bags, because plastic derivatives have much shorter retention times and would elute much earlier in a gas chromatography column. Although alkenones have not been measured in this study, in several years of using this procedure (e.g. Guitián

et al. 2019, Tanner et al. 2020, Guitián et al. 2021), plastic contamination was never observed in the earlier part of the chromatograms during alkenone analysis.

**Sources**

Guitián, J., Phelps, S., Polissar, P. J., Ausín, B., Eglinton, T. I., & Stoll, H. M. (2019). Midlatitude temperature variations in the Oligocene to early Miocene. *Paleoceanography and Paleoclimatology*, *34*(8), 1328-1343.

Guitián, J., & Stoll, H. M. (2021). Evolution of Sea Surface Temperature in the Southern Mid-latitudes From Late Oligocene Through Early Miocene. *Paleoceanography and Paleoclimatology*, *36*(9), e2020PA004199.

Tanner, T., Hernández-Almeida, I., Drury, A. J., Guitián, J., & Stoll, H. (2020). Decreasing atmospheric CO2 during the late Miocene cooling. *Paleoceanography and Paleoclimatology*, *35*(12), e2020PA003925.

Auderset, A., Schmitt, M., & Martínez-García, A. (2020). Simultaneous extraction and chromatographic separation of n-alkanes and alkenones from glycerol dialkyl glycerol tetraethers via selective Accelerated Solvent Extraction. *Organic Geochemistry*, *143*, 103979.

Additional edits:

- Line 34: We have amended these references to better support the point made in the text.
- Line 107: Clarification that the pink crystals were observed in both treated and pre-treated samples.
- Line 207: Included species pair for clarity.
- Line 290-291: Included clarification that percentage difference has been calculated relative to the untreated samples.
- Lines 336-337: Corrected an error where Sr/Ca results were showing as decimal number, not percentages.
- Line 342-343: Provided further clarification for the Mn/Ca results. The greatest offset observed was for the sample pair from 1406B; also included absolute value to demonstrate the size of the spread. All sample pairs are offset compared to the small uncertainty (0.68 %) on Mn/Ca.
- Line 385: Added Hole reference where this was missing.